# An NMR-Based Metabolomics Assessment of the Effect of Combinations of Natural Feed Items on Juvenile Red Drum, *Sciaenops ocellatus*

**Fabio Casu** [1,*] , **David Klett** [1] , **Justin Yost** [1] , **Michael R. Denson** [2] **and Aaron M. Watson** [1,*]

[1] South Carolina Department of Natural Resources, Marine Resources Research Institute, 217 Fort Johnson Road, Charleston, SC 29412, USA; klettd@dnr.sc.gov (D.K.); yostj@dnr.sc.gov (J.Y.)

[2] Hollings Marine Laboratory, National Oceanic and Atmospheric Administration, 331 Fort Johnson Road, Charleston, SC 29412, USA; mike.denson@noaa.gov

*** Correspondence: casuf@dnr.sc.gov (F.C.); watsona@dnr.sc.gov (A.M.W.)

**Abstract:** This study evaluated the effects of seven diets composed of natural feed components (chopped fish, shrimp, and squid) alone or in combination on the liver metabolite profile of juvenile red drum (*Sciaenops ocellatus*) cultured in a 24-tank recirculating aquaculture system over the course of 12 weeks using Nuclear Magnetic Resonance (NMR)-based metabolomics. Experimental diets included fish (F), shrimp (SH), squid (SQ), fish and shrimp (FSH), fish and squid (FSQ), shrimp and squid (SHSQ), fish, shrimp, and squid (FSHSQ). A commercial fishmeal-based pelleted diet was used as a control. Fish were fed isocalorically. Red drum liver samples were collected at five different time points: T0, before the start of the trial ($n$ = 12), and subsequently every 3 weeks over the course of 12 weeks (T3, T6, T9, T12), with $n$ = 9 fish/diet/time point. Polar liver extracts were analyzed by NMR-based metabolomics. Multivariate statistical analyses (PCA, PLS-DA) revealed that red drum fed the F diet had a distinct liver metabolite profile from fish fed the other diets, with those fed SH, SQ and the combination diets displaying greater similarities in their metabolome. Results show that 19 metabolites changed significantly among the different dietary treatments, including amino acids and amino acid derivatives, quaternary amines and methylamines, carbohydrates and phospholipids. Specifically, γ-butyrobetaine, *N*-formimino-L-glutamate (FIGLU), sarcosine and beta-alanine were among the most discriminating metabolites. Significant correlations were found between metabolites and six growth performance parameters (final body weight, total length, condition factor, liver weight, hepatosomatic index, and eviscerated weight). Metabolites identified in this study constitute potential candidates for supplementation in fish feeds for aquaculture and optimization of existing formulations. Additionally, we identified a quaternary amine, γ-butyrobetaine as a potential biomarker of shrimp consumption in red drum. These results warrant further investigation and biomarker validation and have the potential for broader applicability outside of the aquaculture field in future investigations in wild red drum populations and potentially other carnivorous marine fishes.

**Keywords:** aquaculture; metabolomics; NMR; red drum; butyrobetaine; FIGLU

## 1. Introduction

Over the last two decades the global aquaculture industry has seen progressive developments in support of the increasing demand for high-quality, cost-effective, and sustainable seafood production with aquaculture accounting for ~46% of the global fish production and 52% of fish for human consumption in 2018 [1]. In aquaculture, nutrition is of critical importance since costs associated with feeds can account for 50% or more of total production expenses [2]. Recently, a significant body of research has been focusing on the development of new, balanced feed formulations that promote optimal fish growth and health. Complete artificial feeds need to supply all essential nutrients (protein, lipids, carbohydrates), including trace amounts of micronutrients (vitamins and minerals) that are required for

optimal fish growth and health. Additionally, nutritional requirements are species-specific and often depend on other factors such as specific life stage and culture conditions.

Protein constitutes the most expensive component in fish feeds, and specific protein and amino acid requirements may change between species and life stage [3]. Amino acids represent the protein building blocks. Of the 20 naturally occurring amino acids, 10 are essential, and since fish cannot synthesize them, they need to be supplied by the diet. These essential amino acids are arginine, histidine, isoleucine, leucine, lysine, methionine, phenylalanine, threonine, tryptophan, and valine, with lysine and methionine often being the first limiting amino acids [4].

Fishmeal still constitutes a primary protein source for many species in aquaculture; however, with increasing demand and consistent supply from wild fisheries, fishmeal prices have been steadily increasing [1], thus promoting research efforts focused on the evaluation of alternative protein sources for partial or complete fishmeal replacement, specifically plant proteins (e.g., soybean products, cereal grains, legumes) [5,6]. Due to the differences in amino acid profiles between plant proteins and fishmeal (e.g., lower methionine content in soybean meal), plant-based feeds are typically supplemented with several essential nutrients at levels that depend on fish species and life stages to promote optimal growth and health [6–8]. In general, protein requirements are higher for carnivorous fish compared to both herbivorous and omnivorous fish [9–11]. Additionally, smaller fish and early life stage fish have generally higher protein requirements compared with larger fish, since they use the provided protein mostly for growth, when adequate levels of high-energy nutrients (fats and carbohydrates) are supplied in the diet [9,11]. The nutritional value of dietary ingredients depends in part on their ability to supply energy for essential life processes such as digestion, growth, and reproduction, among others [9]. Typically, available energy values for specific nutrients need to be well balanced in formulated feeds. For protein, carbohydrates, and fats, the following average values of caloric content are used in feed formulations: 4, 4, and 9 calories/g, respectively [12,13]. Optimum formulated diets are characterized by a well-balanced energy-to-protein ratio. Any excess or deficiency of supplied energy can have significant effects on fish growth. In fact, an excessive energy supply relative to protein content in feeds can result in high fat deposition in the liver, which can negatively affect fish health, in addition to leading to decreased feed consumption, protein intake, and thus reduced weight gain. On the other hand, when fish are fed an energy-deficient diet relative to protein content, protein is used to satisfy their energy requirements for maintenance before it is available for growth, thus leading to reduced growth rates and weight gain [9,13].

Red drum, *Sciaenops ocellatus*, is a euryhaline marine carnivore with great potential for intensive and extensive culture in the United States. Red drum have been cultured for several decades and general nutritional requirements for red drum are fairly well documented [14–16]. Practical feeds for juvenile red drum generally require ~40% crude protein for optimum growth, in addition to 5–7% crude fat, less than 7% crude fiber, and adequate levels of vitamins and minerals. However, despite the availability of practical feeds for this species, optimal nutritional needs are still under investigation, especially in the context of the evaluation of alternative feed ingredients which are incorporated in increasing amounts in aquafeeds, thus making red drum an excellent warm-water marine model species for nutritional work.

Despite the notable improvements in formulating efficient and cost-effective aquafeeds for different fish species, artificial feeds are still lagging behind natural diets in terms of growth performance, suggesting that the formulated feeds might still be lacking important nutrients, concentrations, or ratios when compared to the natural diet of red drum.

To the best of our knowledge, no other fish nutrition studies published to date have investigated the impacts of natural feed items (fish, shrimp and squid) and various combinations of these components on red drum liver metabolite profiles. This study showcases the application of NMR-based metabolomics to aquaculture nutrition studies which can provide new insights into the existence of specific combinations of dietary components

that promote fish growth and exert positive effects on overall fish health, or otherwise identify suboptimal combinations characterized by the absence, or lower content, of specific metabolites. Specific compounds can be identified that might constitute good candidates for feed supplementation with the potential of enhancing the performance of formulated aquafeeds for red drum and potentially other marine carnivorous species. Additionally, the current study reports on the application of NMR-based metabolomics to the identification of food biomarkers which are indicative of the consumption of specific feed components with potential applicability in future investigations in wild red drum populations and potentially other carnivorous marine fishes.

## 2. Materials and Methods

### 2.1. Experimental Diets

This study evaluated seven experimental diets, which included cut frozen fish (*Decapterus punctatus*), shrimp (*Litopenaeus vannamei*), and squid (*Loligo opalescens* and *Illex*) alone or in combination, in addition to a commercial fishmeal-based pelleted feed that was used as a control throughout the 12-week feeding trial. Frozen feed was purchased from Haddrell's Point Tackle (Charleston, SC, USA). Specifically, the diets included fish only (F), shrimp only (SH), squid only (SQ), fish and shrimp (FSH), fish and squid (FSQ), shrimp and squid (SHSQ), fish, shrimp, and squid (FSHSQ), and a commercial fishmeal-based pelleted feed (PELL). In previous trials conducted in our laboratory on juvenile red drum the diet composed of fish, shrimp, and squid (FSHSQ) ("natural diet") was included as a reference diet for optimal performance, since this diet consistently outperforms commercial and formulated feeds under all growth performance parameters evaluated. Each diet combination was fed to three replicates tanks. A commercial conditioning diet with similar composition to the PELL diet was used during the conditioning period prior to the start of the feeding trial. Based on proximate analysis, the natural feed items (fish, squid, and shrimp) evaluated in this study had a crude protein content (CP) of ~70–80%, while the PELL diet had a CP of ~40%. The crude fat content (CF) was ~10% for the natural items as well as the PELL diet. Additional details on the diets tested in this study are reported elsewhere [17].

### 2.2. Animal Husbandry and Feeding Trial

Wild broodstock red drum underwent volitional spawning at the South Carolina Department of Natural Resources (SCDNR) Marine Resources Research Institute (MRRI) (Charleston, SC, USA). Red drum larvae (single genetic family) were transferred to the Waddell Mariculture Center (WMC) (Bluffton, SC, USA), where they were stocked and grown out to fingerlings into earthen ponds. Fingerlings (30–40 mm average length) were then transported to the Hollings Marine Laboratory (HML) (Charleston, SC, USA) and held in indoor recirculating tanks. Red drum were subsequently distributed in a $24 \times 1100$ L tank recirculating aquaculture system at a stocking density of 25 fish/tank when they reached an average weight of 27.5 g. Tanks are equipped with mechanical and biological filtration systems, UV sterilizers, and temperature control. During a 68-day conditioning period, red drum were fed to apparent satiation twice per day using a commercial pelleted feed (conditioning diet). At the end of the conditioning period, fish were randomly assigned to one of the eight experimental diets and a 12-week feeding trial started. During the feeding trial, red drum were fed isocalorically twice daily. Fish were not fed on sampling days. The amount of feed distributed to each tank (25 fish) was based on average caloric intake and was normalized to the number of calories fed to the fish, shrimp, and squid diet treatment group, so that each tank received the same number of calories at each feeding. Detailed information on husbandry and caloric intake for this study is provided elsewhere [17].

### 2.3. Sample Collection

Sampling was conducted at day 0 (T0) at the beginning of the feeding trial (*n* = 12), and subsequently at weeks 3 (T3), 6 (T6), 9 (T9), and 12 (T12) of the trial (*n* = 9/diet/time

point), for a total of 300 red drum sampled for this study. For liver sample collection, fish were euthanized using a lethal dose (500 mg/L) of tricaine methanesulfonate (MS-222) buffered with sodium bicarbonate for at least 3 min prior to dissection. Fish were then dissected anteriorly from anus to gills and the viscera were removed. The liver was excised, quickly rinsed with a pre-chilled 3% saline solution, transferred into 5 mL cryovials, flash frozen in liquid nitrogen, and stored at −80 °C until further processing for NMR-based metabolomic analysis. Quality control materials used in this study include solvent blanks, replicate experimental samples, a liver pooled control material (LCM) obtained from excess tissue, and NIST Standard Reference Material, SRM 1946 ("Lake Superior Fish Tissue"). Quality control samples were extracted in each batch along with the experimental samples for quality assurance.

### 2.4. Metabolite Extraction for NMR-Based Metabolomics

Frozen liver samples were homogenized individually using a cryomill (Retsch, Inc., Newtown, PA, USA). Sample manipulation during homogenization was performed in a cryogenic cart (Chart Industries, Inc., Garfield Heights, OH, USA) to prevent samples from thawing. Liver homogenates were subsequently aliquoted by weighing 100 mg ($\pm$3 mg) per sample into 2 mL ceramic bead tubes (2.8 mm) (Qiagen, Germantown, MD, USA) for use in a bead beater and stored at −80 °C until extraction. A modified Bligh-Dyer [18–20] solvent extraction protocol was used for metabolite extraction with a final volume ratio chloroform:methanol:water of 2:2:1.8 (*v*:*v*:*v*) as described in detail elsewhere [21–24]. Upon extraction, the polar phase (top layer) was separated using Pasteur pipettes, transferred into microcentrifuge tubes (Eppendorf, Hauppauge, NY, USA) and dried using a vacuum centrifuge (Vacufuge, Eppendorf, Hauppauge, NY, USA) (approximately 2.5 h to 3 h). A total of 600 $\mu$L of NMR buffer (100 mmol/L phosphate buffer in $D_2O$, pH 7.3, with 0.15 mmol/L $NaN_3$ and 1.0 mmol/L sodium 3-(trimethylsilyl)propionate 2,2′,3,3′-*d* (TMSP) as internal chemical shift reference) were added to each sample, vortexed for 30 s, and centrifuged in a bench-top centrifuge for approximately 10 s. In total, 550 $\mu$L of the resulting solution was transferred into 5 mm NMR tubes (Bruker Biospin, Inc., Billerica, MA, USA) for use in a SampleJet automatic sampler (Bruker Biospin) for subsequent NMR analysis.

### 2.5. NMR Spectroscopy Data Acquisition

NMR spectra were acquired at 298 K on a Bruker Avance II 700 MHz spectrometer (Bruker Biospin) equipped with a 5 mm triple-resonance, z-gradient TCI cryoprobe. Specifically, 5 mm NMR tubes were placed in SampleJet 96-well racks (Bruker Biospin) with a 10 min temperature equilibration period before spectral collection. NMR spectra were acquired in automation using ICON-NMR (Bruker Biospin) with automated shimming with on-axis and off-axis shims, automated probe tuning/matching, and pulse calibration. The NMR analysis protocol included a one-dimensional (1D) [1]H NMR experiment with water suppression using the Bruker pulse sequence "noesygppr1d" with a spectral width of 20 ppm, a 3 s relaxation delay, 80 transients, 8 steady-state scans, and 65,536 real data points. A 60 ms mixing period was used for solvent suppression with an acquisition time per experiment of 2.34 s for a total repetition time (D1 + AQ) of 5.34 s. Free induction decays (FIDs were processed by zero-filling to 65,536 complex points and applying an exponential line broadening function (0.3 Hz) prior to Fourier transformation. The resulting NMR spectra were phased, baseline corrected by applying a fifth order polynomial, and chemical shift calibration was performed by setting the standard TMSP peak at 0.00 ppm using Topspin 3.2 (Bruker Biospin). Additional spectra included 2D [1]H-[1]H J-resolved (JRES). When water suppression or linewidth for specific samples was found to be suboptimal, the samples were re-run to improve spectral quality. Two-dimensional (2D) edited [13]C heteronuclear single quantum coherence (HSQC) spectra with adiabatic [13]C decoupling (hsqcedetgpsisp2.3) with 25% non-uniform sampling (NUS) were collected on selected samples for metabolite identification. NUS schedules were generated using the default seed value of the random number generator implemented within Topspin 3.2. The number

of complex points in the direct dimension was 2048 and 64 complex points were acquired in the indirect dimension with spectral widths of 11 ppm in F2 and 180 ppm in F1 ($^{13}$C). The number of scans was 256. A relaxation delay equal to 1.5 s was used between acquisitions, and a refocusing delay corresponding to a 145 Hz 1$J_{C-H}$ coupling was used. The FIDs were weighted using a shifted sine-square function in both dimensions. NUS spectra were reconstructed using iterative soft thresholding according to the hmsIST algorithm applied as a compressed-sensing (CS) approach implemented in Topspin 4.0. All spectra were referenced to the TMSP internal standard at 0.00 ppm for $^1$H and $^{13}$C.

### 2.6. NMR Data Analysis and Multivariate Statistical Data Analysis

Metabolites found to be statistically significant were identified based on the comparison of experimental chemical shifts and coupling constants with those available in databases such as the Human Metabolome Database (HMDB, http://www.hmdb.ca, accessed 23 December 2021) [25], the Biological Magnetic Resonance Data Bank (BMRB, http://bmrb.wisc.edu/, accessed 23 December 2021) [26], the Birmingham Metabolite Library for 2D $^1$H-$^1$H JRES spectra [27], and an in-house compiled database as well as tables found in published reports [28]. In general, metabolite identification was achieved at a Level 2, putative identification level [29]. A limited number of spectral features that could not be identified based on database or literature searches were annotated as "Unknown". For multivariate statistical analysis, the pre-processed 1D $^1$H NMR spectra were binned (bin size: 0.005 ppm) in the spectral region between δ 10.0 ppm and 0.2 ppm. The following spectral regions were excluded from the analysis since they contained artifacts either due to water suppression or the presence of contaminants detected in blank spectra: acetate (1.93–1.91 ppm), water (4.90–4.70 ppm), chloroform (7.69–7.67 ppm), and formate (8.47–8.45 ppm). Spectral alignment and binning were performed using NMRProcFlow 1.3 software (www.nmrprocflow.org, accessed 23 September 2021). Prior to multivariate analysis, the binned NMR spectra were normalized to the sum of total spectral intensities and Pareto scaling was applied. Data normalization, scaling, and multivariate analysis were performed using MetaboAnalyst 5.0 software (www.metaboanalyst.ca, accessed 30 November 2021). Multivariate statistical analysis included principal component analysis (PCA) and partial least squares discriminant analysis (PLS-DA). PCA score plots allowed the identification of specific clusters within the dataset. PLS-DA Variable Importance in Projection (VIP) scores >1.0 were used to select spectral features (metabolites) that were the most discriminating among the dietary treatments. For the purpose of quality assurance, NMR spectra obtained from QC samples were assessed for spectral relative standard deviation (RSD = standard deviation/mean × 100%) [30] to evaluate the potential variation associated with our experimental protocol. RSD values < 10% for LCM and SRM samples were considered an indication of good experimental repeatability.

### 2.7. Pathway Analysis

Metabolites found to be significantly different among the different dietary treatments were mapped using the Pathway Analysis module implemented in MetaboAnalyst 5.0 software (www.metaboanalyst.ca, accessed 10 March 2022). The zebrafish (*Danio rerio*) pathway library was utilized, and hypergeometric test and relative betweenness centrality algorithms were used. For each pathway analyzed, a fit coefficient (*p*) was calculated from pathway enrichment analysis and an impact factor was obtained from pathway topology analysis.

### 2.8. Statistical Analysis

PCA score plots displaying average score values per group are presented as mean ± SEM. Clustering in these PCA score plots was assessed for significant differences using Student's *t*-tests (two-tailed, unequal variance) in Microsoft Excel (Microsoft Corporation, Redmond, WA, USA). Differences were considered statistically significant if the *p*-value was less than 0.05. One-way analysis of variance (ANOVA) was performed to test the

significance of differences in metabolite levels among different treatments. Post hoc comparisons of the means were conducted using Tukey's multiple comparison test. Multivariate statistical calculations (PCA and PLS-DA) were performed using MetaboAnalyst 5.0. For PLS-DA, $Q^2$ (predictive ability of the model), $R^2$ (goodness of fit), and the *p*-value of the permutation test (1000 permutations) were assessed. Models were accepted as valid for $Q^2 > 0.5$ and $p < 0.05$. Spearman's correlations between metabolite levels and growth performance parameters were performed using GraphPad Prism version 8 (GraphPad Software, Inc., La Jolla, CA, USA).

## 3. Results

### 3.1. Red Drum Growth Performance Indices

Results from the growth performance are described elsewhere [17]. Briefly, at the end of the 12-week feeding trial significant differences were detected in SGR, final body weight, final total length, condition factor, FCR, and HSI (ANOVA, $p < 0.05$) among red drum fed the different diets. Specifically, the F diet was associated with the highest SGR, average final weight, average final length, and condition factor. In contrast, fish fed diets SH, SQ, and SHSQ had the lowest SGR, average final weight, average final length, and condition factor. As far as FCR, the F diet was associated with the lowest FCR suggesting that this diet was more efficient, whereas fish fed diets SH, SQ, and SHSQ had the highest FCR, along with fish fed the pelleted control diet (PELL). Additionally, red drum fed PELL were characterized by significantly higher liver weight and HSI values, followed by fish fed the F diet compared with all other diets.

### 3.2. Comparison of Liver Metabolite Profiles

Representative 1D $^1$H NMR spectra of juvenile red drum liver extracts from fish fed the seven experimental diets and the commercial pelleted diet are shown (Supplementary Material, Figure S1). Initially, principal component analysis was performed on the pre-processed 1D $^1$H NMR spectra to assess the effect of the different diets on red drum liver metabolite profiles and assess data clustering. Analysis of QC samples was assessed for spectral RSD with values for LCM and SRM samples of 7.3% and 6.7%, respectively, which are indicative of good experimental repeatability (Supplementary Material, Figure S2). The resulting PCA score plots for the first 3 principal components (PC1, PC2, and PC3) are shown in Figure 1. A clear separation along PC1 was observed between the fish fed the F diet and the fish fed all other diets. PC1 explained 35.8% of the total variance within the data. Additionally, fish fed the conditioning diet (T0) and the PELL diet were not significantly different from each other, but a significant separation was observed between fish fed these diets and all other diets along PC2, which explained 23.9% of the total variance. Fish fed the SH diet, the SQ diet, and any combination of the natural feed items showed a high degree of similarity based on the proximity of the respective clusters in the PCA score plots.

To evaluate metabolic fingerprints that were characteristic of specific dietary treatments and identify potential biomarkers of consumption of any of these feed items, we proceeded with performing partial least squares discriminant analysis (PLS-DA), a supervised multivariate statistical method that utilizes the class membership information for the different groups to maximize group separation and correlate observed differences with specific metabolites. The resulting PLS-DA score plots are shown (Figure 2). Permutation testing showed the model to be valid ($p < 0.001$).

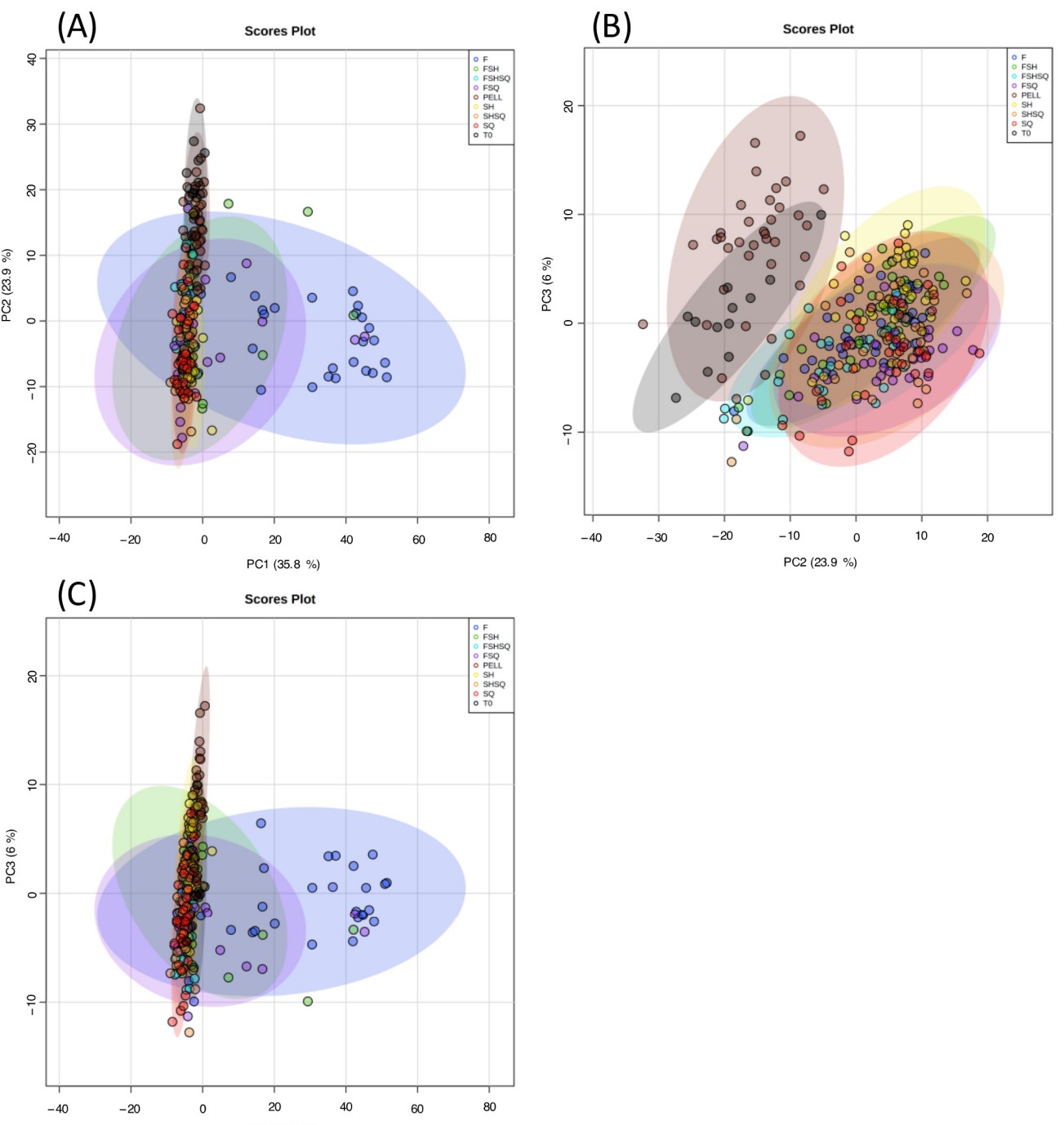

**Figure 1.** Unsupervised PCA score plots from red drum liver tissue extracts for the 8 diets evaluated in this study and the conditioning diet (T0). (**A**) PC1/PC2 score plot. (**B**) PC2/PC3 score plot. (**C**) PC1/PC3 score plot. The explained variances along the different principal components are shown in brackets. Shaded areas represent the 95% confidence regions. T0, initial time point (conditioning diet, same composition as PELL); F, fish diet; FSH, fish + shrimp diet; FSHSQ, fish + shrimp + squid diet; FSQ, fish + squid diet; PELL, pelleted commercial fishmeal-based diet; SH, shrimp diet; SHSQ, shrimp + squid diet; SQ, squid diet.

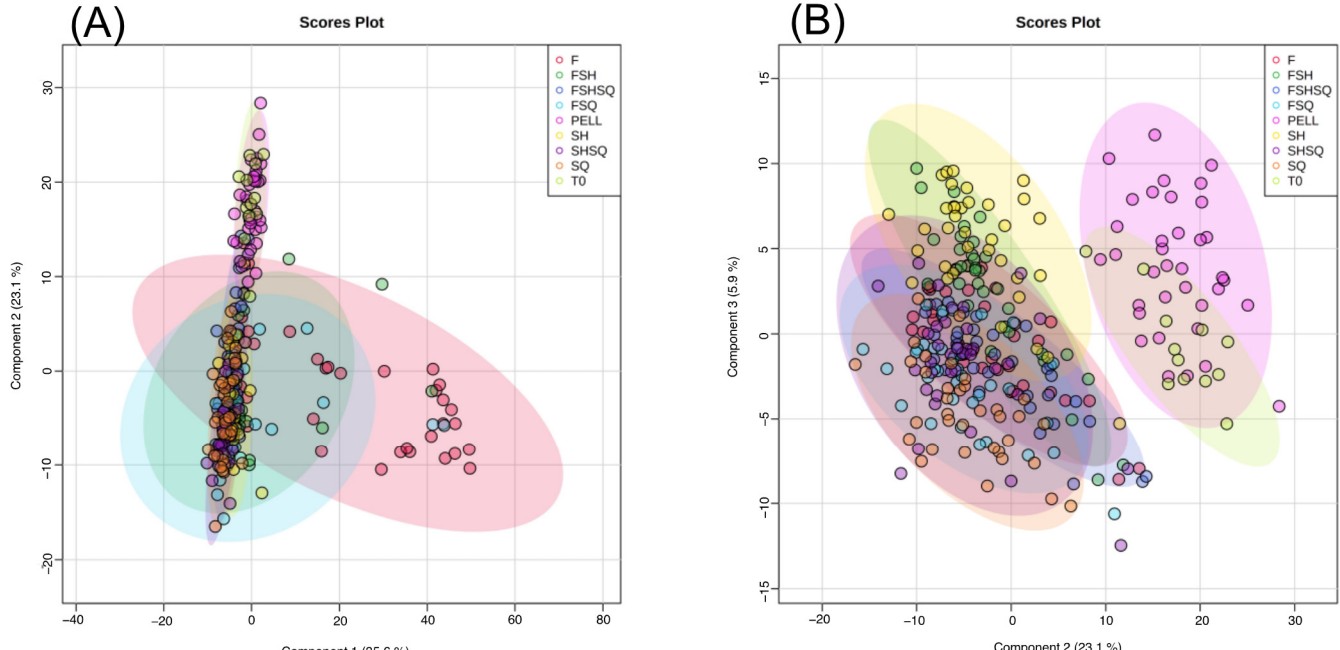

**Figure 2.** Supervised PLS-DA score plot from red drum liver tissue extracts for the 8 diets evaluated in this study and the conditioning diet (T0). (**A**) PC1/PC2 score plot. (**B**) PC2/PC3 score plot. The explained variances along the principal components are shown in brackets. Shaded areas represent the 95% confidence regions. PLS-DA score plot model validation (NC = 2; $R^2$ = 0.455; $Q^2$ = 0.430; 1000 permutations: $p < 0.001$).

A total of 18 metabolites and 1 unknown compound were found to be significantly different among the nine diets (including the conditioning diet, T0), with VIP scores > 1.0 (Table 1).

**Table 1.** VIP scores for the 19 most discriminating metabolites among the dietary treatments identified from PLS-DA of $^1$H NMR spectra of red drum liver extracts.

| Bin (ppm) | VIP Score | Metabolite |
|---|---|---|
| 3.2675 | 5.5331 | TMAO |
| 3.4275 | 4.9697 | Taurine |
| 7.8325 | 4.9104 | FIGLU |
| 3.2725 | 2.9042 | Betaine |
| 3.3075 | 1.9947 | Proline-betaine |
| 3.4075 | 1.9541 | Glucose |
| 3.1975 | 1.8755 | Taurine-betaine |
| 2.3525 | 1.8423 | Glutamate |
| 3.1375 | 1.8218 | γ-Butyrobetaine |
| 3.3875 | 1.7702 | Unkn_105 |
| 3.8375 | 1.7455 | Glycerol 3-phosphate |
| 2.7425 | 1.5722 | Sarcosine |
| 3.5625 | 1.3904 | Glycine |
| 3.2275 | 1.3840 | O-phosphocholine |
| 2.1475 | 1.3109 | Glutamine |
| 2.5575 | 1.1444 | beta-Alanine |
| 1.4875 | 1.1146 | Alanine |
| 5.2325 | 1.0883 | Glucose 6-phosphate |
| 3.2075 | 1.0608 | Cystathionine |

Relative metabolite levels for the 19 metabolites are shown as a heatmap in Figure 3.

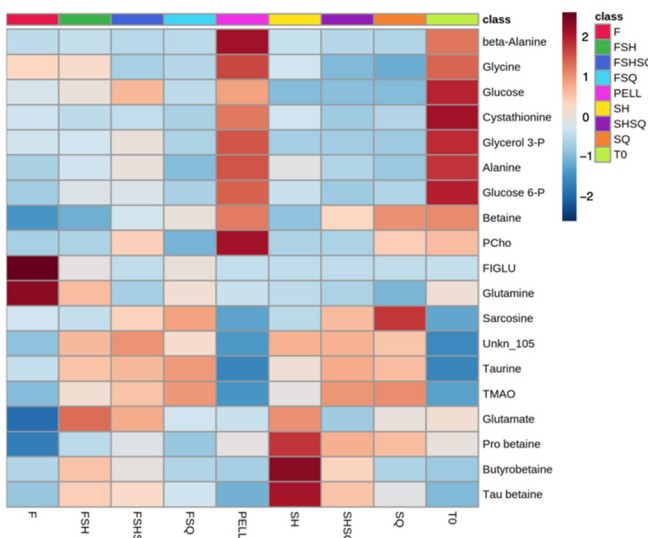

**Figure 3.** Heat map showing relative metabolite levels for the 19 most discriminating metabolites among the 8 diets (F, FSH, FSHSQ, FSQ, PELL, SH, SHSQ, SQ) and the conditioning diet (T0) measured in red drum liver extracts. A gradient color scale from red (high) to blue (low) indicates mean relative levels of the 19 most discriminating metabolites (rows) in the 9 diets (columns). F, fish diet; FSH, fish + shrimp diet; FSHSQ, fish + shrimp + squid diet; FSQ, fish + squid diet; PELL, pelleted commercial fishmeal-based diet; SH, shrimp diet; SHSQ, shrimp + squid diet; SQ, squid diet. Glycerol 3-P, glycerol 3-phosphate; glucose 6-P, glucose 6-phosphate; PCho, O-phosphocholine; FIGLU, *N*-formimino-L-glutamate; Pro betaine, proline-betaine; Tau betaine, taurine-betaine.

Red drum fed the F diet were characterized by the highest hepatic levels of FIGLU and glutamine, and lowest levels of betaine, glutamate, and proline-betaine among all the diets; fish fed the SH diet were characterized by the highest levels of γ-butyrobetaine, glutamate, proline-betaine, and taurine-betaine; and fish fed the SQ diet were characterized by the highest levels of sarcosine, taurine, and TMAO, and lowest levels of alanine, cystathionine, glucose, and glutamine. Overall, fish fed combinations of these feed items (fish, shrimp, and squid) showed intermediate levels of these metabolites. Specifically, red drum fed the FSH diet showed elevated hepatic levels of glutamate, glutamine, taurine, γ-butyrobetaine, taurine-betaine, and an unknown compound (unknown_105), but low levels of betaine. Fish fed the FSQ diet displayed elevated levels of sarcosine, taurine, and TMAO, but low levels of O-phosphocholine, alanine, and proline-betaine. Red drum fed the SHSQ diet showed elevated levels of TMAO, sarcosine, taurine, proline-betaine, taurine-betaine, betaine, γ-butyrobetaine, and an unknown compound (unknown_105), but low levels of glycine, glucose, cystathionine, glycerol 3-P, alanine, glucose 6-P, and glutamate. Fish fed the FSHSQ diet were characterized by elevated levels of glucose, betaine, glutamate, sarcosine, taurine, and TMAO, taurine-betaine, and an unknown compound (unknown_105), but low levels of glycine and glutamine. Finally, red drum fed the PELL diet and the conditioning diet (T0) had similar metabolite profiles, and specifically they were characterized by significantly higher levels of alanine, beta-alanine, betaine, cystathionine, glucose, glucose 6-phosphate, glycerol 3-phosphate, and O-phosphocholine, and lower levels of sarcosine, taurine, TMAO, and taurine-betaine.

Relative metabolite levels of FIGLU, beta-alanine, γ-butyrobetaine, and sarcosine, among the most discriminating metabolites for the F, SH, SQ, and PELL (and conditioning diet, T0) are also shown for all dietary treatments (Figure 4).

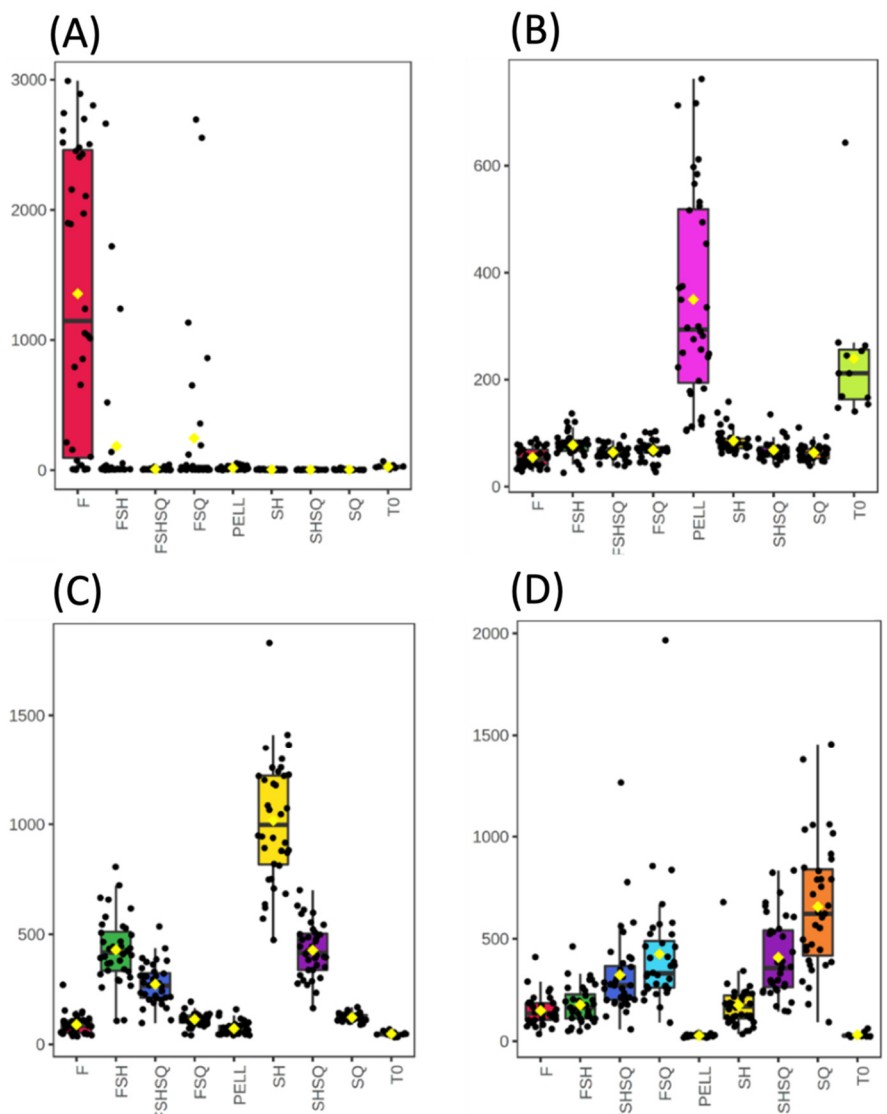

**Figure 4.** Box plots showing relative metabolite levels among the 8 diets (F, FSH, FSHSQ, FSQ, PELL, SH, SHSQ, SQ) and the conditioning diet (T0) measured in red drum liver extracts. (**A**) FIGLU; (**B**) beta-alanine; (**C**) γ-butyrobetaine; (**D**) sarcosine. F, fish diet; FSH, fish + shrimp diet; FSHSQ, fish + shrimp + squid diet; FSQ, fish + squid diet; PELL, pelleted commercial fishmeal-based diet; SH, shrimp diet; SHSQ, shrimp + squid diet; SQ, squid diet.

*3.3. Metabolomic Time Trajectories*

To evaluate potential time-dependent changes in metabolite profiles over the course of the 12-week feeding trial, the same PC1–PC2 score plot was simplified by displaying the mean scores for the five different sampling points (T0, T3, T6, T9, T12) from liver samples collected at T0 (at the start of the feeding trial) or every 3 weeks after the start of the feeding trial (T3, T6, T9, T12) for all the eight diets (Figure 5).

All diets at T3 with the exception of the PELL diet were significantly different from T0 along PC2 ($p < 0.05$), or both along PC1 and PC2 in the case of the F diet at T6, T9, and T12 ($p < 0.05$).

When considering the single-component diets (F, SH, and SQ), fish fed the F diet showed significant differences ($p = 0.03$) between T3 and T6 samples along PC1; however, no significant differences were found among fish fed the F diet at T6, T9, or T12 along PC1 or PC2 ($p > 0.05$). Fish fed the SH diet displayed significant differences ($p = 0.02$) between T9 and T12 samples along PC2, but not along PC1; no significant differences were detected

among fish fed the SH diet when comparing the other time points either along PC1 or PC2 ($p > 0.05$). Significant differences ($p < 0.05$) were found in fish fed the SQ diet between T3 samples and those collected at the other time points (T6, T9, and T12) along PC1 but not along PC2. No significant differences were found between fish fed the SQ diet when comparing the other time points either along PC1 or PC2 ($p > 0.05$).

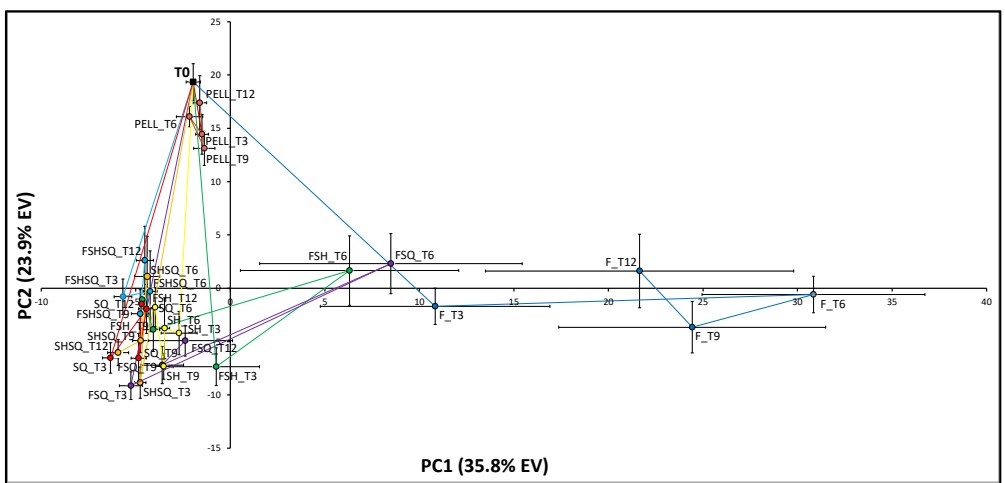

**Figure 5.** Unsupervised PCA score plot (mean ± SEM) from red drum liver tissue extracts for the 8 diets at 5 different time points (T0, T3, T6, T9, T12) over the course of the 12-week red drum feeding trial. The explained variances along PC1 and PC2 are shown in brackets. For each time point, the mean scores are indicated with circles or a square (T0); SEM error bars are displayed both in PC1 and PC2 for each time point. T0, initial time point (conditioning diet, same composition as PELL); F, fish diet; FSH, fish + shrimp diet; FSHSQ, fish + shrimp + squid diet; FSQ, fish + squid diet; PELL, pelleted commercial fishmeal-based diet; SH, shrimp diet; SHSQ, shrimp + squid diet; SQ, squid diet.

As far as the combination diets (FSH, FSQ, SHSQ, and FSHSQ), fish fed the FSH diet displayed significant differences between T3 and both T6 ($p = 0.03$) and T12 ($p = 0.04$) along PC2 but not along PC1. No other significant differences were found between fish fed the FSH diet when comparing the other time points either along PC1 or PC2 ($p > 0.05$). Significant differences were detected in fish fed the FSQ diet between T3 and both T6 ($p < 0.01$) and T12 ($p = 0.04$) along PC2 but not along PC1. Significant differences were also detected in fish fed the FSQ diet between T6 and both T9 ($p = 0.01$) and T12 ($p = 0.04$) along PC2 but not along PC1. No significant differences were found between fish fed the FSQ diet when comparing T9 and T12 either along PC1 or PC2 ($p > 0.05$). Both FSH and FSQ diets at T6 were not significantly different from fish fed the F diet at T3, T9, and T12 ($p > 0.05$) both along PC1 and PC2, but they were different from fish fed the F diet at T6 ($p > 0.05$) along PC1 but not along PC2. Fish fed the SHSQ diet showed significant differences between T3 and T6 ($p = 0.03$) along PC2 but not along PC1. Additionally, significant differences were detected between T3 and T9 along PC1 ($p < 0.01$) but not along PC2. No other significant differences were found between fish fed the SHSQ diet when comparing the other time points either along PC1 or PC2 ($p > 0.05$). Finally, in the case of fish fed the FSHSQ and those fed the PELL diet, there were no significant differences between the different time points either along PC1 or PC2.

### 3.4. Correlations between Metabolites and Growth Performance Indices

Correlations between metabolite levels (bin intensities) for the 19 identified metabolites and 6 fish growth performance metrics (body weight, total length, liver weight, eviscerated weight, hepatosomatic index (HSI), and condition factor) were calculated using the Spearman's correlation coefficient (non-parametric) for all dietary treatments (Figure 6 and Table 2).

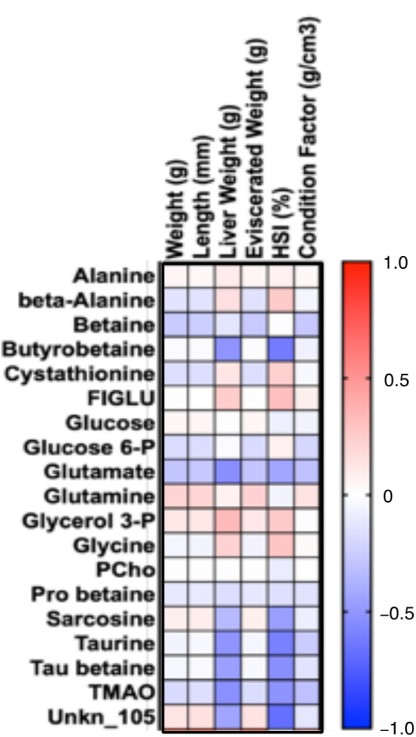

**Figure 6.** Spearman's correlation matrix showing correlation of metabolite levels of the 19 most discriminating metabolites with 6 performance indices (body weight, total length, liver weight, eviscerated weight, hepatosomatic index (HSI), and condition factor). A gradient color scale from red (positive correlation) to blue (negative correlation) indicates values of Spearman's *rho* correlation coefficients.

**Table 2.** Spearman's *rho* correlation coefficients of the 19 most discriminating metabolites with 6 performance indices (body weight, total length, liver weight, eviscerated weight, hepatosomatic index (HSI), and condition factor).

| Liver Metabolite | Body Weight (g) | Total Length (mm) | Liver Weight (g) | Eviscerated Weight (g) | HSI (%) | Condition Factor (g/cm³) |
|---|---|---|---|---|---|---|
| Alanine | 0.042 | 0.037 | 0.099 | 0.046 | 0.072 | 0.036 |
| *beta*-Alanine | −0.137 | −0.144 | 0.156 | −0.146 | 0.259 | −0.053 |
| Betaine | −0.260 | −0.245 | −0.137 | −0.266 | −0.008 | −0.271 |
| γ-Butyrobetaine | −0.026 | −0.024 | −0.515 | −0.017 | −0.627 | −0.077 |
| Cystathionine | −0.161 | −0.169 | 0.125 | −0.168 | 0.230 | −0.038 |
| FIGLU | 0.005 | −0.001 | 0.248 | 0.001 | 0.308 | 0.080 |
| Glucose | 0.045 | 0.049 | 0.009 | 0.050 | −0.085 | −0.063 |
| Glucose 6-phosphate | −0.174 | −0.165 | −0.025 | −0.185 | 0.072 | −0.200 |
| Glutamate | −0.290 | −0.275 | −0.537 | −0.287 | −0.433 | −0.310 |
| Glutamine | 0.210 | 0.208 | 0.070 | 0.218 | −0.062 | 0.145 |
| Glycerol 3-phosphate | 0.117 | 0.122 | 0.333 | 0.116 | 0.262 | −0.013 |
| Glycine | −0.053 | −0.062 | 0.221 | −0.062 | 0.279 | 0.028 |
| O-phosphocholine | 0.010 | 0.005 | −0.014 | 0.009 | −0.098 | −0.006 |
| Proline-betaine | −0.117 | −0.109 | −0.165 | −0.121 | −0.160 | −0.142 |
| Sarcosine | 0.077 | 0.091 | −0.338 | 0.086 | −0.469 | −0.090 |
| Taurine | −0.062 | −0.039 | −0.517 | −0.053 | −0.602 | −0.257 |
| Taurine-betaine | −0.046 | −0.033 | −0.465 | −0.039 | −0.540 | −0.141 |
| TMAO | −0.185 | −0.163 | −0.533 | −0.180 | −0.519 | −0.307 |
| Unkn_105 | 0.130 | 0.149 | −0.438 | 0.143 | −0.675 | −0.126 |

As a rule of thumb, the correlation was considered "very strong" for absolute values of the Spearman's *rho* coefficient 0.80–1.0, "strong" for values 0.60–0.79, "moderate" for values 0.40–0.59, "weak" for values 0.20–0.39, and "very weak" for values <0.19.

Glutamine showed weak positive correlations with body weight (0.21), total length (0.21), and eviscerated weight (0.22), while glutamate showed weak negative correlations with body weight ($-0.29$), total length ($-0.27$), eviscerated weight ($-0.29$), and condition factor ($-0.31$). Betaine showed weak negative correlations with body weight ($-0.26$), total length ($-0.24$), eviscerated weight ($-0.27$), and condition factor ($-0.27$). Taurine and TMAO were negatively correlated with condition factor ($-0.26$ and$-0.31$, respectively). A number of metabolites showed significant correlations with liver weight and HSI. FIGLU, glycerol 3-phosphate, glycine, beta-alanine, and cystathionine all showed weak positive correlations with both liver weight and HSI, with FIGLU showing the highest positive correlation (0.31). On the other hand, $\gamma$-butyrobetaine, taurine and an unknown compound (unknown_105) showed a moderate negative correlation with liver weight, and a strong negative correlation with HSI. Glutamate, taurine-betaine, and TMAO all showed moderate negative correlations with both liver weight and HSI, while sarcosine displayed a weak negative correlation with liver weight ($-0.34$) and a moderate negative correlation with HSI ($-0.47$).

### 3.5. Pathway Analysis

Pathway analysis revealed that amino acid metabolism was the most impacted by the different dietary treatments in red drum liver with glycine, serine, and threonine metabolism showing the highest value of $-\log(p)$, followed by glutamine and glutamate metabolism, and alanine, aspartate, and glutamate metabolism, arginine biosynthesis, and histidine metabolism. Glycerophospholipid metabolism was also affected, as well as glutathione metabolism and taurine metabolism (Supplementary Material, Figure S4).

## 4. Discussion

This study was designed to test the effects of seven different combinations of natural feed components which included cut fish (*Decapterus punctatus*), shrimp (*Litopenaeus vannamei*), and squid (*Loligo opalescens* and *Illex*), on the liver metabolite profiles of juvenile red drum cultured in a flow-through recirculating aquaculture system over the course of a 12-week feeding trial. In a number of feeding trials conducted by our research team on juvenile red drum, a "natural" diet composed of cut fish, shrimp, and squid was included as a reference diet for optimal performance [21]. In all trials conducted in our laboratory the "natural" reference diet significantly and consistently outperforms commercial and formulated feeds containing alternative protein sources (e.g., soybean meal or soy protein concentrate) under all growth performance parameters evaluated [21]. In a previous feeding experiment conducted by our research team, on average juvenile red drum consumed an amount of feed corresponding to 8.4% of their body weight when fed the fish, shrimp, and squid diet to satiation (unpublished data). In this study red drum were fed to 7.8% of their body weight to ensure complete feed consumption, similar to other studies conducted on red drum juveniles [31–33]. Additionally, in this feeding trial, fish feeding was standardized based on caloric intake (0.07 cal/g fish/day), and due to the different caloric content of the individual dietary components (fish, shrimp and squid), red drum were fed different total amounts (grams) of each dietary component.

Among all diets tested in this study, fish fed the F diet formed a cluster in PCA plots that was significantly separated from fish fed all other diets, thus indicating that they were characterized by a distinct metabolite profile (Figure 1). Additionally, fish fed the PELL diet had a similar metabolite profile to those fed the conditioning diet as indicated by the significant degree of overlapping between the two groups in the corresponding PCA score plots, which is consistent with the two diets having similar compositions. The PELL/conditioning diet fish cluster showed a clear separation from red drum fed all other diets, which is to be expected based on the differences in composition between the pelleted

diets and the natural feed components. Ultimately, fish fed the SH and SQ diets and the various combination diets (FSH, FSQ, SHSQ, FSHSQ) showed a higher degree of similarity as indicated by the significant overlap of the corresponding samples in the PCA score plots. PLS-DA score plots of the same samples revealed significant differences in metabolite profiles between fish fed the SH diet and those fed the SQ diet, despite their higher degree of similarity when compared to fish fed the F diet or the PELL/conditioning diet (Figure 2).

The evaluation of metabolomic trajectories over the course of the 12-week feeding trial revealed some time-dependent differences when comparing fish at the five different sampling time points (T0, T3, T6, T9, T12) (Figure 5). Fish metabolite profiles at T0 were found to be significantly different from those at the other time points; however, the significant degree of overlapping observed among all time points in the case of the PELL diet (with minor differences detected only for T9) seems to suggest that differences between T0 and the other time points were mostly associated with a switch in diet composition rather than a time-dependent modulation of the metabolome. With the exception of fish fed the F diet and those fed the two-component diets containing fish (FSH, FSQ), a significant degree of overlapping was observed among the different time points (T3 through T12), suggesting that overall red drum liver metabolite profiles did not change significantly after the first 3 weeks following the initial dietary switch at T0 to the natural diet components and combinations.

In order to correlate the observed separation of the various dietary treatments with differences in the relative levels of specific metabolites detected in the $^1$H NMR spectra, we performed PLS-DA. Using PLS-DA, we identified a total of 18 metabolites and 1 unknown compound that were responsible for the observed differences in metabolite profiles among the 9 diets (including the conditioning diet, T0) tested in this study (Figure 3 and Table 1). Among these metabolites are compounds that have been previously shown to stimulate feeding behavior in different fish species, such as the amino acids alanine and glycine, the non-proteinogenic amino acid taurine and quaternary amines such as betaine and TMAO [34–36]. The fish diet was the most significantly different from all other diets, with this group separation being responsible for most of the variance along PC1 in the PCA score plots, while a higher degree of similarity was observed among diets containing shrimp and squid. This observation is consistent with a more similar chemical composition between different marine invertebrates (e.g., molluscs and crustaceans), compared with marine teleost fishes in agreement with previous studies [35]. PLS-DA revealed that this separation correlated with significantly higher levels of both FIGLU and glutamine along with lower levels of glutamate and proline-betaine in the F diet compared with the other diets. Specifically, FIGLU ranked as the most discriminating hepatic metabolite for the F diet based on VIP scores. A time-course evaluation of average FIGLU levels per time point revealed that in fish fed the F diet, FIGLU levels increased progressively from T0 to T3 and subsequently from T3 to T6, reaching a maximum at T6, with levels slightly decreasing between T9 and T12 (Supplementary Material, Figure S3). Similarly, fish fed the two-component diets containing fish (FSH, FSQ), also showed an increase in FIGLU levels from T0 to T6; however, FIGLU levels were significantly lower compared with average FIGLU levels at each time point in fish fed the F diet. Changes in FIGLU levels over time are primarily responsible for the observed differences along PC1 in the PCA score plot for fish fed the F, FSH, and FSQ diets (Figure 5).

*N*-formimino-L-glutamate (FIGLU) is a metabolic intermediate of histidine catabolism [37,38]. FIGLU is normally converted into glutamate by a folate-dependent enzyme, unless there is a deficiency in folate, vitamin B12 (enzyme cofactor) or methionine which can prevent its metabolism and cause FIGLU accumulation in the liver and excretion in the urine [38–44]. FIGLU has been utilized as a biomarker of folate and vitamin B12 deficiency in mammals since the late 1950s when the urine "FIGLU test" was developed for folate and vitamin B12 deficiency detection in humans [38–44]. Since histidine, the precursor of FIGLU, is an essential amino acid in mammals, fish, and poultry because it cannot be synthesized endogenously, it needs to be obtained from the diet. An imbalance between histidine

input and available folate, vitamin B12 or methionine from dietary sources can lead to significant increases in hepatic FIGLU levels. Analysis of the amino acid composition of different species of fish, molluscs, and crustaceans have shown that among the essential amino acids, histidine levels were significantly higher in fish samples compared with both mollusc (including squid) and crustacean (including shrimp) samples, with squid showing the lowest levels overall [45]. Methionine levels varied significantly in different species of molluscs and crustaceans, but they were significantly lower in shrimp (*Penaeus* sp.) compared with fish samples [45]. The overall lower content in these essential amino acids found in squid and shrimp might be responsible at least in part for the significantly lower performance of the corresponding diets (SQ, SH) compared with the fish diet (F) observed in this juvenile red drum feeding study.

Correlations of metabolites with growth performance parameters revealed that considering all dietary treatments and time points, FIGLU showed a weak positive correlation with HSI and liver weight, but was not correlated with fish body weight, total length, condition factor, or eviscerated weight (Figure 6 and Table 2). Overall, with the exception of fish fed the PELL/conditioning diet, fish with higher HSI values were characterized by higher hepatic FIGLU levels, which could indicate altered hepatic metabolism in fish with increased lipid deposition in the liver.

Among the amino acids found to be the most discriminating among the different diets tested in this study, with the exception of alanine, that did not show significant correlations with any of the performance metrics evaluated, glutamate, glutamine, and glycine showed either positive or negative correlations with body weight, total length, condition factor, eviscerated weight, liver weight, or HSI. Specifically, glutamine showed the strongest positive correlation with fish body weight, total length, condition factor, and eviscerated weight, while glutamate displayed the strongest negative correlation with the same parameters in addition to liver weight among the 19 metabolites. While both glutamine and glutamate are considered non-essential amino acids in fish nutrition, they are known to play fundamental roles particularly in intestinal health by improving intestinal structure and functionality, promoting innate and adaptive immune responses as well as providing protection against oxidative damage. Previous studies have shown that glutamine supplementation can promote fish growth in different species, including red drum [46–48].

Another amino acid, the non-proteinogenic amino acid beta-alanine, was identified as one of the most discriminating metabolites among the different diets. Beta-alanine is a non-essential naturally occurring β-amino acid present particularly in white and red meat. Other than from dietary sources, beta-alanine can be synthesized endogenously in the liver [49]. Primarily, beta-alanine is generated from the degradation of uracil [50]. Importantly, beta-alanine is a component of natural histidine-containing dipeptides such as anserine and carnosine, which play an important role in the pH homeostasis of muscles during exercise [50]. Additionally, beta-alanine is a substrate for pantothenic acid (vitamin B5) biosynthesis, which in turn is a component of coenzyme-A and is important in the metabolism of lipids (e.g., fatty acids, cholesterol, steroid hormones), carbohydrates, and also protein [51]. Based on our results, beta-alanine levels showed a very weak positive correlation with liver weight and a weak positive correlation with HSI.

Pathway analysis performed using the 18 identified metabolites shows that overall, amino acid metabolism was the most impacted by the different dietary treatments (Supplementary Material, Figure S4).

In addition to amino acids, a number of quaternary amines and methylamines were important in discriminating between red drum fed the different diets, among which we identified trimethylamine *N*-oxide (TMAO) and the betaines glycine-betaine (betaine), proline-betaine, taurine-betaine, and γ-butyrobetaine. TMAO was found to be significantly higher in the liver tissue of fish fed the diets containing squid (SQ, SHSQ, FSQ, FSHSQ) compared with those fed the F diet and the PELL diet (and conditioning diet) in agreement with previous reports showing high levels of TMAO in invertebrates compared with teleost

fish [52]. Glycine-betaine (betaine) is known to act as an osmoprotectant, by protecting the cells from dehydration, high salinity, temperatures, and osmotic stress [53]. Betaine, and its precursor choline, can be provided with the diet, with higher levels of betaine generally found in mollusc and crustacean tissues, and lower levels present in teleost species [35]. Betaine supplementation of fish feeds, especially those containing fishmeal alternatives such as plant-based protein sources, have shown positive effects in both freshwater and marine fish species, with an overall increase in feed intake and growth suggesting betaine can significantly improve feed palatability [54–56]. Betaine levels were found to be significantly higher in the liver of fish fed the SQ diet and the PELL diet (and conditioning diet) compared with those fed the F diet and SH diet.

In relation to the betaine metabolism, we also identified sarcosine as a metabolite with high discriminating power among the dietary treatments. Sarcosine is an intermediate of the choline and betaine metabolism, which is primarily synthesized in the liver and kidney [57]. Fish fed the SQ diet showed the highest levels of sarcosine in the liver tissue, whereas those fed the PELL (and conditioning) diet had the lowest hepatic levels of this metabolite. Since sarcosine is generated from the betaine metabolism, and betaine levels were found to be higher in fish fed the SQ diet, this can explain, at least in part, the elevated levels of sarcosine observed in these fish. However, fish fed the PELL and conditioning diets, which also displayed high betaine levels, were characterized by significantly lower sarcosine levels compared with those fed the SQ diet, which could be the result of an increased rate of metabolic degradation of sarcosine to glycine in these fish.

In addition to betaine, γ-butyrobetaine was found to be significantly different in different dietary treatments. We found γ-butyrobetaine to be a potential marker of shrimp consumption in red drum with the highest levels detected in the liver tissue of fish fed the SH diet, followed by those fed FSH and SHSQ, and then FSHSQ compared with those fed diets that did not contain shrimp (Figure 4). Based on other reports, γ-butyrobetaine has not been detected or has been detected only at low concentrations in fish, molluscs, and crustaceans [58,59]. The high levels of γ-butyrobetaine in fish fed the SH diet can therefore be attributed to enhanced biosynthesis rather than dietary input. γ-Butyrobetaine is a known precursor of L-carnitine, which in turn plays an important role in the regulation of fatty acid (FA) metabolism [60]. In fact, L-carnitine is involved in the transport of long-chain fatty acids (LCFAs) from the cytoplasm to the mitochondrial matrix for degradation via β-oxidation, which is important in energy production [61]. In all animals including fish, L-carnitine can be provided by dietary sources, but it can also be synthesized endogenously in some tissues (primarily the kidneys, liver, and brain) from the amino acids lysine and methionine [62,63]. The last step in L-carnitine biosynthesis involves the formation of L-carnitine from γ-butyrobetaine, in a reaction dependent on α-ketoglutarate [64]. Dietary supplementation of L-carnitine has been associated with decreased lipid accumulation and overall improved the lipid metabolism in individuals exposed to high-fat diets (HFD) or those affected by certain metabolic disorders such as non-alcoholic fatty liver disease (NAFLD) in humans and other animals; however, the results obtained in fish studies have so far been inconsistent [65–70]. Similar to L-carnitine, its precursor γ-butyrobetaine has been investigated for its potential positive effects on juvenile visceral steatosis in mice [71].

Our results showed that γ-butyrobetaine, taurine-betaine, TMAO, and an unknown compound (unknown_105) were all negatively correlated with both liver weight and HSI, with unkn_105 showing the strongest negative correlation with HSI (−0.67), followed by γ-butyrobetaine (−0.63). Furthermore, betaine showed negative correlations with body weight, total length, condition factor, and eviscerated weight. Proline-betaine did not show significant correlations with any of the performance metrics evaluated in this study.

Another metabolite associated with lipid metabolism, glycerol 3-phosphate, was found to be higher in fish fed the PELL/conditioning diet compared with fish fed the natural feed components (fish, shrimp, and squid) in agreement with previous reports [21]. Glycerol 3-phosphate is a known substrate of mitochondrial glycerol 3-phosphate dehydrogenase which is at the interface between the lipid and carbohydrate metabolism [72,73]. We found

that glycerol 3-phosphate showed a weak positive correlation with both liver weight (0.33) and HSI (0.26). Although, there were no significant differences in crude fat (CF) content between the PELL diet (~10%) and the natural feed components in terms of dry mass, the PELL group was fed significantly more dry feed per fish per day, due to the different composition of the pelleted feed and the natural feed items and therefore fish were fed higher levels of crude fat over the course of the trial. Fish fed the PELL diet showed significantly higher HSI values compared to all the natural diets, which could also be the result of a different lipid composition and/or digestibility of the pelleted feed compared with the natural feed items [17].

O-phosphocholine, a choline derivative and an intermediate in the synthesis/degradation of phosphatidylcholine (lecithin), constitutes the main structural phospholipid in eukariotic biomembranes [74]. O-phosphocholine was detected at higher levels in fish fed the PELL/conditioning diet compared with fish fed the natural feed components (fish, shrimp, and squid), which suggests modulation of membrane lipid turnover and lipid redistribution to peripheral tissues, likely due to significant differences in lipid composition between terrestrial feed ingredients in manufactured feeds and marine ingredients [73]. Pathway analysis confirms that glycerophospholipid metabolism was among the most impacted metabolic pathways by the different dietary treatments (Supplementary Material Figure S4).

Metabolites associated with energy metabolism, specifically glucose metabolism (glucose, glucose 6-phosphate), were found to be significantly different between the PELL/conditioning diet and the other diets, with higher levels of these compounds measured in fish fed the PELL and conditioning diet compared with all other diets, suggesting modulation of energy metabolism in fish fed these feeds compared with natural feed components (fish, shrimp, and squid), results that are consistent with previous studies on formulated feeds in juvenile red drum [21,75]. It is important to highlight the fact that while the CF content was not significantly different, the PELL diet had a significantly lower protein content (CP: ~40%) compared with the natural feed items (CP: ~70–75%). The differences in protein content and obviously water content (~70–80% for natural feed items) and therefore "nutrient density" may explain some of the differences observed between red drum fed the PELL diet and fish fed the natural feed components such as differences related to the energy metabolism. In terms of growth performance, the PELL feed performed second best after the F diet under a caloric restriction regime, which indicates that, despite the different composition, this diet is nutritionally balanced overall.

Additional metabolites found to be significant in discriminating among the different diets were taurine and cystathionine, an intermediate of cysteine metabolism and taurine biosynthesis. Taurine is a non-proteinogenic sulfur amino acid that has been shown to be essential for optimal growth of several commercially important species, especially marine teleosts [76–79]. In fact, most fish species are not capable of synthesizing taurine, or sufficient taurine for optimal growth, endogenously, and therefore taurine needs to be supplied through the diet either with the inclusion of marine ingredients with high taurine content (e.g., fishmeal, squid meal, etc.) or via direct taurine supplementation [77,78]. Taurine has been shown to modulate the amino acid, carbohydrate, lipid, and nucleotide metabolism as well as immune responses, osmoregulation, and displaying antioxidant properties [77,78,80]. Taurine content in marine invertebrates has generally been observed to be higher than in marine teleosts [35,58,81,82]. In agreement with these reports, our results show that taurine levels were higher in fish fed diets containing squid (SQ, SHSQ, FSQ, FSHSQ, as well as the combination diet with fish + shrimp (FSH), with the highest levels measured in fish fed the FSQ diet.

## 5. Conclusions

Despite the significant improvements in formulating efficient feeds for different fish species, manufactured feeds continue to lag behind natural diets in terms of growth performance, suggesting that the formulated feeds might still be lacking important nutrients when compared to the natural diets especially for carnivorous species such as red drum.

In this study, we used an NMR-based metabolomics approach to evaluate the effects of different combinations of natural feed components (fish, shrimp, and squid) on the liver metabolite profiles of juvenile red drum over the course of a 12-week feeding trial. Using multivariate statistical analyses (PCA, PLS-DA), we were able to identify a total of 18 metabolites and 1 unknown compound that were the most discriminating among the different dietary treatments. Pathway analysis revealed that amino acid metabolism and glycerophospholipid metabolism were among the metabolic pathways most impacted by the dietary treatments in the liver of juvenile red drum. The non-essential amino acid glutamine was positively correlated with growth performance in agreement with its use as a feed supplement to improve fish growth in various fish species. Additionally, a number of metabolites including taurine, γ-butyrobetaine, taurine-betaine, and TMAO were negatively correlated with liver weight and HSI, the supplementation of which could potentially improve the effects of increased HSI due to diet-induced alteration of lipid metabolism and subsequent lipid deposition in the liver. The lower performance of feed items such as squid and shrimp appear to be due at least in part to their lower content in some essential nutrients, particularly EAAs such as histidine and methionine. Ultimately, of particular interest is the identification of γ-butyrobetaine as a potential biomarker of shrimp consumption in red drum, which warrants further investigation and validation and has the potential for applicability in future investigations in wild red drum populations and potentially other carnivorous marine fishes. γ-Butyrobetaine and other metabolites identified in this study constitute potential candidates for supplementation in fish feeds and should be further evaluated.

**Supplementary Materials:** The following supporting information can be downloaded at: https://www.mdpi.com/article/10.3390/jmse10040547/s1, Figure S1: Representative 1D $^1$H NMR spectra of juvenile red drum liver extracts for fish fed the 7 experimental diets and the commercial pelleted diet; Figure S2: Liver QC samples PCA score plot; Figure S3: Relative hepatic FIGLU levels (bin intensity, 7.83 ppm) measured over the course of the 12-week juvenile red drum feeding trial at T0, T3, T6, T9, and T12; Figure S4: Metabolomic pathway analysis overview showing the metabolic pathways that are mostly impacted by the different dietary treatments.

**Author Contributions:** Conceptualization, D.K., J.Y., M.R.D. and A.M.W.; Data curation, F.C.; Formal analysis, F.C.; Funding acquisition, D.K. and A.M.W.; Investigation, F.C. and D.K.; Project administration, A.M.W.; Validation, F.C.; Writing—original draft, F.C.; Writing—review and editing, F.C., M.R.D. and A.M.W. All authors have read and agreed to the published version of the manuscript.

**Funding:** This work was funded in part by a Slocum Lunz Foundation Grant to D. K. and Saltwater Recreational Fisheries Advisory Committee (SRFAC) funds.

**Institutional Review Board Statement:** The animal study protocol was approved by the Institutional Animal Care and Use Committee of the College of Charleston (IACUC Approval 2020-002).

**Informed Consent Statement:** Not applicable.

**Data Availability Statement:** All data generated or analyzed during this study are included in this article.

**Acknowledgments:** The authors would like to thank SCDNR personnel Gabrielle Fignar, Mary Ann Taylor, and Maggie Jamison for their assistance with fish tissue sampling, and Molly Milstein for assistance with tissue processing. This work is contribution number 851 from the South Carolina Department of Natural Resources, Marine Resources Research Institute.

**Conflicts of Interest:** The authors declare no conflict of interest.

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
