# Peer review of "An NMR-Based Metabolomics Assessment of the Effect of Combinations of Natural Feed Items on Juvenile Red Drum, Sciaenops ocellatus"

_jmse, doi:10.3390/jmse10040547_

Round 1
Reviewer 1 Report
The description of the NMR experiment, although correct, may be confusing to a reader unfamiliar with the NMR technique. The NOESY technique is intended to study interactions between close atoms in space, using direct couplings between atoms (NOESY = Nuclear Overhauser Effect Spectroscopy). In this experiment, one signal is selectively irradiated and the change in intensity of other signals is observed. However, the sequence related to the NOESY experiment can serve as a tool for solvent suppression (this is a case of the submitted work). In fact, the authors did not carry out the 1H NOESY experiment with water suppression (noesygppr1d), but rather the 1H NMR experiment with water suppression using the Bruker pulse sequence "noesygppr1d". Consequently, in the Supporting Information section, the authors present a set of 1H NMR spectra with water signal suppressed, not a set of NOESY spectra.
Author Response
Point 1: “The description of the NMR experiment, although correct, may be confusing to a reader unfamiliar with the NMR technique. The NOESY technique is intended to study interactions between close atoms in space, using direct couplings between atoms (NOESY = Nuclear Overhauser Effect Spectroscopy). In this experiment, one signal is selectively irradiated and the change in intensity of other signals is observed. However, the sequence related to the NOESY experiment can serve as a tool for solvent suppression (this is a case of the submitted work). In fact, the authors did not carry out the 1H NOESY experiment with water suppression (noesygppr1d), but rather the 1H NMR experiment with water suppression using the Bruker pulse sequence "noesygppr1d". Consequently, in the Supporting Information section, the authors present a set of 1H NMR spectra with water signal suppressed, not a set of NOESY spectra.”
Response 1: The authors agree with the reviewer and corrected the instances where 1H NOESY NMR experiments were mentioned in the main text and the Supporting Information. To reflect these changes:
In the manuscript main text:
- Lines 205-206 were changed from “The NMR analysis protocol included one-dimensional (1D) 1H NOESY experiments with water suppression (noesygppr1d)…” to “The NMR analysis protocol included a one-dimensional (1D) 1H NMR experiment with water suppression using the Bruker pulse sequence “noesygppr1d”.
- Lines 238-239 were changed from “the pre-processed 1D 1H NOESY NMR spectra…” to “the pre-processed 1D 1H NMR spectra”.
- Line 293 was changed from “Representative 1D 1H NOESY NMR spectra …” to “Representative 1D 1H NMR spectra …”.
- Line 296 was changed from “the pre-processed 1D 1H NOESY NMR spectra …” to “the pre-processed 1D 1H NMR spectra …”.
- Table 1 title was changed from “PLS-DA of 1H NOESY NMR spectra …” to “PLS-DA of 1H NMR spectra…”.
- Line 538 was changed from “detected in the 1H NOESY NMR spectra …” to “detected in the 1H NMR spectra …”.
- Line 761 was changed from “Representative 1D 1H NOESY NMR spectra …” to “Representative 1D 1H NMR spectra …”.
In the Supporting Information:
The caption to Figure S1 was changed from “Representative 1D 1H NOESY NMR spectra…” to “Representative 1D 1H NMR spectra with water signal suppressed using the Bruker pulse sequence “noesygppr1d”…”.
Reviewer 2 Report
This paper contains an attempt to explain in detail the differences in liver constituents among different diets using statistical methods and this study is a significant attempt to clarify the nutritional requirements of the target fish. However, one of the fundamental flaws of this study is that it lacks essential data on the composition and proportions of the ingredients in the diet. The intake of these components naturally affects the liver composition. Although the feeding amounts are to be published in another paper, this data is also essential for drawing the conclusions of this paper, as it also has a significant effect on liver composition.
Therefore, if the composition of the feed changes, all the data in this paper will change, and the reader will not be able to refer to the contents of this paper. Therefore, reviewer concluded that the contents of this paper alone do not provide meaningful new scientific findings.
Other
The letters and symbols in the figures are small and difficult to distinguish even when enlarged. Design innovations are desirable.
Author Response
Point 1: “However, one of the fundamental flaws of this study is that it lacks essential data on the composition and proportions of the ingredients in the diet. The intake of these components naturally affects the liver composition. Although the feeding amounts are to be published in another paper, this data is also essential for drawing the conclusions of this paper, as it also has a significant effect on liver composition.”, “Therefore, if the composition of the feed changes, all the data in this paper will change, and the reader will not be able to refer to the contents of this paper. Therefore, reviewer concluded that the contents of this paper alone do not provide meaningful new scientific findings.”
Response 1: In response to the reviewer’s comment, the authors would like to clarify a fundamental point about the diets evaluated in this study. As mentioned in the Materials and Methods section and throughout the manuscript, the experimental diets were cut up natural feed items (fish, squid and shrimp) and not formulated, pelleted feeds. The proportions of these in the combination diets were based on caloric content for each feed item. As far as the commercial feeds used in this study (conditioning feed and control feed), they are closed formulations.
Proximate composition for the diets used in this study is reported in a recent publication cited in the manuscript (Klett, D.A., Watson, A.M. Aquaculture Nutrition, 2022). However, information has been added to the Materials and Methods section, at the end of subsection 2.1. Experimental Diets of this manuscript for clarity, as follows: “Based on proximate analysis, the natural feed items (fish, squid and shrimp) evaluated in this study had a crude protein content (CP) of ~70-80%, while the PELL diet had a CP of ~40%. The crude fat content (CF) was ~ 10% for the natural items as well as the PELL diet. Additional details on the diets tested in this study are reported elsewhere (Klett, D.A., Watson, A.M. Aquaculture Nutrition, 2022)”.
Point 2: “The letters and symbols in the figures are small and difficult to distinguish even when enlarged. Design innovations are desirable”
Response 2: The authors improved the quality of the following figures:
- Figure 1: we rearranged panels A, B and C by placing panel C under A and B, to improve readability of letters and symbols in the figure.
- Figure 3: we increased the size of Figure 3 to improve readability of the letters and symbols.
- Figure 4: we increased the size of Figure 4 to improve readability of the letters and symbols.
Reviewer 3 Report
The Authors of the manuscript titled "An NMR-based metabolomics assessment of the effect of combinations of natural feed items in juvenile red drum, Sciaenops ocellatus" have presented an interesting and well-written paper, which will be a good addition to the field of aquaculture. Unfortunately, while the majority of the text is of supreme quality, there is a crucial problem within the Materials and Methods section and this paper cannot be published unless this issue is resolved.
I have aligned my commentary in a paragraph by paragraph manner, to be seen below.
Abstract: Everything in this section is correct, all necessary information is clearly outlined. I want only to point towards the erroneous font change in Lines 28-30.
Introduction: To be honest, this is an excellently written overview of the study's topic, easy to read and to comprehend. However, my main issue here is that the entire paragraph found in Lines 98-118 belongs to the M&M section, as it mostly describes the concept of the feeding study and also explains the choices made by the Authors by outlining the details of their previous work. Some of that information containing references might actually be split and moved into the Discussion instead. Nevertheless, after insertion into the M&Ms, some parts of this paragraph should be reworked to be a better fit with whatever is already indicated in that section.
Materials and Methods: Apart from the missing information which needs to be moved here from the Introduction, most of the technical description appears to be spot on, especially given how lengthy it is. However, I spotted one major problem within this section: the lack of any details about the formulated feeds! This is inacceptable. A table containing the % or g/kg amounts of specific ingredients, as well as the proximate composition of these diets is obligatory for this type of studies. How can someone conduct such precise research on different protein sources without actually indicating at least their inclusion levels within the diets? Furthermore, there is also no information about the commercial diet, which also needs to be mentioned as it is the only point of reference. Please provide all the necessary data.
Results: The written part of this section is well-versed and comprehensively presents the obtained scientific data. However, I found some issues in regard to the figures and tables, listed below.
Figure 1: I believe these plots could be bigger, the legend is barely visible. Please try to rework them to the sizes withing Figure 2, likely by moving plot C beneath A and B.
Table 1: I am quite sure MDPI will demand to rework this Table in accordance with their own template.
Figure 4: Same commentary as above - please improve the size of the plots.
Table 4: Same commentary as above - please use the correct template.
Discussion: Given that the previous section is truly long, there was a lot of information which needed to be commented on, but the Authors certainly managed to touch upon all of the results, what once again outlines their high competences in this area. However, with the aforementioned lacking information about diets, the presented commentary feels to be incomplete. Thus, I highly recommend to include this missing information, as it severely hinders the overall perception of the mansucript.
Conclusions: I believe this part is slightly too long. I suggest to narrow it down to the most crucial elements and final outlines/possibilities for future studies.
Author Response
Point 1: “The Authors of the manuscript titled "An NMR-based metabolomics assessment of the effect of combinations of natural feed items in juvenile red drum, Sciaenops ocellatus" have presented an interesting and well-written paper, which will be a good addition to the field of aquaculture. Unfortunately, while the majority of the text is of supreme quality, there is a crucial problem within the Materials and Methods section and this paper cannot be published unless this issue is resolved.”
Response 1: See Point 4 - “Materials and Methods” comments and response below.
Point 2: “Abstract: Everything in this section is correct, all necessary information is clearly outlined. I want only to point towards the erroneous font change in Lines 28-30.”
Response 2: We corrected the erroneous font type in the abstract, Lines 28-30.
Point 3: “Introduction: To be honest, this is an excellently written overview of the study's topic, easy to read and to comprehend. However, my main issue here is that the entire paragraph found in Lines 98-118 belongs to the M&M section, as it mostly describes the concept of the feeding study and also explains the choices made by the Authors by outlining the details of their previous work. Some of that information containing references might actually be split and moved into the Discussion instead. Nevertheless, after insertion into the M&Ms, some parts of this paragraph should be reworked to be a better fit with whatever is already indicated in that section.”
Response 3: As suggested by the reviewer the authors moved some of the information from Lines 98-118 to the Materials and Methods section and to the Discussion. Specifically:
- In the Materials and Methods, subsection 2.1. Experimental Diets, Lines 138-139 were changed from “in addition to a commercial fishmeal-based pelleted feed that was used as a control.” To “in addition to a commercial fishmeal-based pelleted feed that was used as a controlthroughout the 12-week feeding trial.
- In the Materials and Methods, subsection 2.1. Experimental Diets, the following text was added to Line 142 “In previous trials conducted in our laboratory on juvenile red drum the diet composed of fish, shrimp and squid (FSHSQ) (“natural diet”) was included as a reference diet for optimal performance, since this diet consistently outperforms commercial and formulated feeds under all growth performance parameters evaluated.”
- The following paragraph containing references was moved to the beginning of the Discussion section: This study was designed to test the effects of seven different combinations of natural feed components which included cut fish (Decapterus punctatus), shrimp (Litopenaeus vannamei), and squid (Loligo opalescens and Illex), on the liver metabolite profiles of juvenile red drum cultured in a flow-through recirculating aquaculture system over the course of a 12-week feeding trial. In a number of feeding trials conducted by our research team on juvenile red drum, a “natural” diet composed of cut fish, shrimp and squid was included as a reference diet for optimal performance [17]. In all trials conducted in our laboratory this “natural” reference diet significantly and consistently outperforms commercial and formulated feeds regardless of protein sources (g., fishmeal, soybean meal or soy protein concentrate) under all growth performance parameters evaluated [17]. In a previous feeding experiment conducted by our research team, on average juvenile red drum consumed an amount of feed corresponding to 8.4% of their body weight when fed the fish, shrimp, and squid diet to satiation (unpublished data). In this study red drum were fed to 7.8% of their body weight to ensure complete feed consumption, similar to other studies conducted on red drum juveniles [18–20], as well as to force any potential dietary deficiencies to be displayed when they may otherwise be masked in fish able to feed to satiation. Additionally, in this feeding trial fish feeding was standardized based on caloric intake (0.07 cal/g fish/day), and due to the different caloric content of the individual dietary components (fish, shrimp and squid), red drum were fed different total amounts (grams) of each dietary component.
Point 4: “Materials and Methods: Apart from the missing information which needs to be moved here from the Introduction, most of the technical description appears to be spot on, especially given how lengthy it is. However, I spotted one major problem within this section: the lack of any details about the formulated feeds! This is inacceptable. A table containing the % or g/kg amounts of specific ingredients, as well as the proximate composition of these diets is obligatory for this type of studies. How can someone conduct such precise research on different protein sources without actually indicating at least their inclusion levels within the diets? Furthermore, there is also no information about the commercial diet, which also needs to be mentioned as it is the only point of reference. Please provide all the necessary data.”
Response 4: In response to the reviewer’s comment, the authors would like to clarify a fundamental point about the diets evaluated in this study. As mentioned in the Materials and Methods section and throughout the manuscript, the experimental diets were cut up natural feed items (fish, squid and shrimp) and not formulated, pelleted feeds. As mentioned in the manuscript, the proportions of these in the combination diets were based on caloric content for each feed item. As far as the commercial feeds used in this study (conditioning feed and control feed), they are closed formulations. Proximate composition for the diets used in this study is reported in a recent publication cited in the manuscript (Klett, D.A., Watson, A.M. Aquaculture Nutrition, 2022). However, information has been added to the Materials and Methods section, at the end of subsection 2.1. Experimental Diets of this manuscript for clarity, as follows: “Based on proximate analysis, the natural feed items (fish, squid and shrimp) evaluated in this study had a crude protein content (CP) of ~70-80%, while the PELL diet had a CP of ~40%. The crude fat content (CF) was ~ 10% for the natural items as well as the PELL diet. Additional details on the diets tested in this study are reported elsewhere (Klett, D.A., Watson, A.M. Aquaculture Nutrition, 2022)”.
Point 5: “Results: The written part of this section is well-versed and comprehensively presents the obtained scientific data. However, I found some issues in regard to the figures and tables, listed below.
Figure 1: I believe these plots could be bigger, the legend is barely visible. Please try to rework them to the sizes withing Figure 2, likely by moving plot C beneath A and B.
Response 5: As suggested by the reviewer, the authors rearranged panels A, B and C in Figure 1 by placing panel C under A and B, to increase the size of the plots and corresponding legends.
Point 6: Table 1: I am quite sure MDPI will demand to rework this Table in accordance with their own template.
Response 6: As suggested by the reviewer, the authors formatted Table 1 according to MDPI guidelines.
Point 7: Figure 4: Same commentary as above - please improve the size of the plots.
Response 7: As suggested by the reviewer, the authors increased the size of Figure 4 to improve readability.
Point 8: Table 4: Same commentary as above - please use the correct template.”
Response 8: As suggested by the reviewer, the authors formatted Table 2 according to MDPI guidelines.
Point 9: “Discussion: Given that the previous section is truly long, there was a lot of information which needed to be commented on, but the Authors certainly managed to touch upon all of the results, what once again outlines their high competences in this area. However, with the aforementioned lacking information about diets, the presented commentary feels to be incomplete. Thus, I highly recommend to include this missing information, as it severely hinders the overall perception of the mansucript.”
Response 9: See Response 4.
Point 10: “Conclusions: I believe this part is slightly too long. I suggest to narrow it down to the most crucial elements and final outlines/possibilities for future studies.”
Response 10: As suggested by the reviewer, the authors worked on making the Conclusions more concise. Specifically:
- The following text from Lines 730-732 was removed: “Notable progress has been made in aquaculture nutrition with the need for more sustainable cost-effective feed formulations, which often include alternative ingredients to common nutrient sources such as fishmeal. However…”
- The following text from Lines 741-743 was removed: “These metabolites include amino acids and amino acid derivatives, quaternary amines and methylamines, carbohydrates and phospholipids.”
- Line 745 was changed from “…pathways that were most impacted by the dietary treatments” to “pathways most impacted by the dietary treatments”
- Lines 745-748 were changed from “We identified glutamine as being a metabolite that was positively correlated with growth performance in agreement with its use as a feed supplement to improve fish growth in various fish species, in addition to a number of metabolites…” to “The non-essential amino acid glutamine was positively correlated with growth performance in agreement with its use as a feed supplement to improve fish growth in various fish species. Additionally, a number of metabolites…”
Round 2
Reviewer 2 Report
The composition of the bait was clarified and the problem was solved.
Reviewer 3 Report
The Authors of the newly resubmitted manuscript titled "An NMR-based metabolomics assessment of the effect of combinations of natural feed items in juvenile red drum, Sciaenops ocellatus" have introduced all necessary corrections into their paper, following the guidelines from the first round of review.
The aforementioned paragraph was fittingly removed from the "Introduction" and split into the "Materials and Methods" and "Discussion" sections, which made the whole text much clearer to follow and logical. I also believe the Authors have successfully addressed my complaints about the lack of any commentary about the studied diets. I admit that I might have previously missed the information about "frozen" fish, shrimp and squid being given to the fish instead of pelleted feed, but this was partially a consequence of this lacking part of the study design. Currently, I am satisfied with the outcome, although I would still strongly suggest the Authors to add the Table with the basic composition of the diets, which is Table 3 in their other paper [17].
In the "Results" section, all Figures and Tables were improved in quality, as demanded. Meanwhile, the shortened "Conclusions" are now adequate.
All in all, what had to be changed has indeed been changed. Well done!